# Study protocol for transforming health equity research in integrated primary care: Antiracism as a disruptive innovation

Sylvie Naar[1]*, Carrie Pettus[2], Norman Anderson[3], Meardith Pooler-Burgess[1], Penny Ralston[4], Heather Flynn[5], Todd Combs[6], Claudia Baquet[7], Christopher Schatschneider[8], Douglas Luke[6]

1 Center for Translational Behavioral Science, Florida State University, Tallahassee, FL, United States of America, 2 Wellbeing & Equity Innovations, Tallahassee, FL, United States of America, 3 Office of Vice President for Research and College of Social Work, Florida State University, Tallahassee, FL, United States of America, 4 Center on Better Health and Life for Underserved Populations, Florida State University, Tallahassee, FL, United States of America, 5 Department of Behavioral Sciences and Social Medicine, Florida State University, Tallahassee, FL, United States of America, 6 Center for Public Health Systems Science, Brown School, Washington University in St. Louis, St. Louis, Missouri, United States of America, 7 Hope Institute, LLC and UM School of Pharmacy, Baltimore, Maryland, United States of America, 8 Department of Psychology, Florida State University, Tallahassee, FL, United States of America

* sylvie.naar@med.fsu.edu

**Data Availability Statement:** Deidentified research data will be made publicly available when the study is completed and published.

## Abstract

Among the consequences of systemic racism in health care are significant health disparities among Black/African American individuals with comorbid physical and mental health conditions. Despite decades of studies acknowledging health disparities based on race, significant change has not occurred. There are shockingly few evidence-based antiracism interventions. New paradigms are needed to intervene on, and not just document, racism in health care systems. We are developing a transformative paradigm for new antiracism interventions for primary care settings that integrate mental and physical health care. The paradigm is the first of its kind to integrate community-based participatory research and systems science, within an established model of early phase translation to rigorously define new antiracism interventions. This protocol will use a novel application of systems sciences by combining the qualitative systems sciences methods (group model building; GMB) with quantitative methods (simulation modeling) to develop a comprehensive and community-engaged view of both the drivers of racism and the potential impact of antiracism interventions. Community participants from two integrated primary health care systems will engage in group GMB workshops with researchers to 1) Describe and map the complex dynamic systems driving racism in health care practices, 2) Identify leverage points for disruptive antiracism interventions, policies and practices, and 3) Review and prioritize a list of possible intervention strategies. Advisory committees will provide feedback on the design of GMB procedures, screen potential intervention components for impact, feasibility, and acceptability, and identify gaps for further exploration. Simulation models will be generated based on contextual factors and provider/patient characteristics. Using Item Response Theory, we will initiate the process of developing core measures for assessing the effectiveness of interventions at the organizational-systems and provider levels to be tested under a variety of

**Funding:** The study has received financial grant support from the National Institute of Health's Transformative Research Awards (1R01MD017404). The funders have had no role in the design, implementation, analysis or write up of the study.

**Competing interests:** The authors have no competing interests.

conditions. While we focus on Black/African Americans, we hope that the resulting transformative paradigm can be applied to improve health equity among other marginalized groups.

## Introduction

The significant health disparities in the prevalence, diagnosis, and treatment of comorbid physical and mental health conditions among Black/African Americans has been well established [1–5]. These inequities synergistically result in poor physical and mental health outcomes, including increased mortality in Black/African Americans [6,7]. Integrating mental health and primary health care for underinsured populations, as in the patient-centered medical home model, was expected to reduce these disparities [8–10]. Yet, such hopes have not been fully realized. Disparities persist and access and engagement in integrated primary care are still inadequate among persons of color.

Such disparities can, in part, be explained by racism in health care [11–13]. Unequal treatment of racial minorities across multiple levels of the health care system will not be eradicated by structural interventions to improve social determinants of health [14]. Racism in health care settings towards Black/African Americans in particular has been recently declared a public health emergency [15]. The literature has captured a great deal of evidence about racism among health care providers in the US. In one systematic review alone, 26 out of 37 studies showed "statistically significant evidence of racist beliefs." An assessment of implicit and explicit bias against persons of color among primary care providers and community members revealed substantial implicit bias against both Latinos and Black/African Americans among both providers and community members, while explicit bias was absent or difficult to detect for both [16]. Not surprisingly, clinicians' implicit bias negatively affects patients' perceptions of their care and reluctance to engage in primary care and follow-up with recommendations [12,17–19].

There is currently a strong need to develop antiracism interventions, which are those that are explicitly designed to eliminate racism in health care settings. Calliste, Dei and Sefa [20] defined antiracism interventions as an "action-oriented, educational and/or political strategy for systemic and political change that addresses issues of racism and interlocking systems of social oppression." Such interventions may take the form of "individual transformation, organizational change, community change, movement-building, anti-discrimination legislation and racial equity policies in health, social, legal, economic and political institutions" [20].

Several systematic reviews capture the current state of interventions to reduce racism in health care settings, but few interventions are grounded in antiracism specifically and instead focus on prejudices, stereotypes, cultural competency, and cultural appropriateness. These are, of course, ways of targeting racism, albeit in a more indirect way. One review covered 30 peer-reviewed studies conducted between 2005 and 2015 on interventions designed to reduce implicit prejudices and implicit stereotypes in real world contexts. Few studies reported robust data, and many interventions showed no effect or actually increased implicit bias [21]. In a systematic review of reviews of articles published between 2000 and 2012 on interventions to improve cultural competency in health care, 19 published reviews were identified. Among the conclusions of the review of reviews: many of the studies relied on self-report, many studies lacked methodological rigor, and few studies included evidence of intervention effectiveness [22]. Finally, in a systematic review to establish whether cultural competency training of health professionals improves patient outcomes the conclusion is similar to other studies: High-

quality research on interventions designed to improve cultural competency and reduce various race-related biases is lacking. Similarly, the literature does not often contain studies revealing a positive relationship between cultural competency training and improved patient outcomes [23] and instead shows limited impact. Based on these reviews, it is clear that more targeted work to develop and implement antiracism interventions in health care settings is needed, and demand for such interventions is high and necessary to achieve health equity.

Only one study to date has leveraged systems science to address structural racism in community health [24]. The result was community driven systems maps to leverage key points and identify solutions for system change in a single county; however, interventions were not specified. Racism in health care is woven throughout a complex system that can be disrupted by diffusing antiracism innovations at critical points in the system. A new transformative paradigm is needed to ignite the field of antiracism health care interventions to change behavior, policies and practices in order to fully realize the potential of integrated primary care models to improve equity in physical and mental health for Black/African Americans. An authentic community-engaged approach necessitates the focus on one primary ethnic group while considering intersectionality [25]. However, if successful, such a paradigm could be translated to develop antiracism strategies to improve health equity among other marginalized populations.

Funded by the National Institute of Health Director's Transformative Research Award, this project aims to transform this field by combining scientifically proven social, behavioral, and systems science approaches and applying them to change behavior, policies, and practices of individual providers and organizational systems. The purpose of this paper is to describe the study protocol and present a transformative model for community engaged systems science to develop new antiracism interventions.

## Theoretical approaches

A transformative approach to complex problems requires the integration of several models into an overarching transformative framework. Because most models are not explicitly antiracist, we begin with the Social Ecological Model (SEM) model of antiracism which holds that antiracism interventions must consider racism occurring at different levels, including the individual, interpersonal, community, organizational, and policy levels [26,27].

Second, we will use the Obesity-related Behavioral Intervention Trials (ORBIT) model of translational behavioral and social science (see Fig 1), an established phased intervention

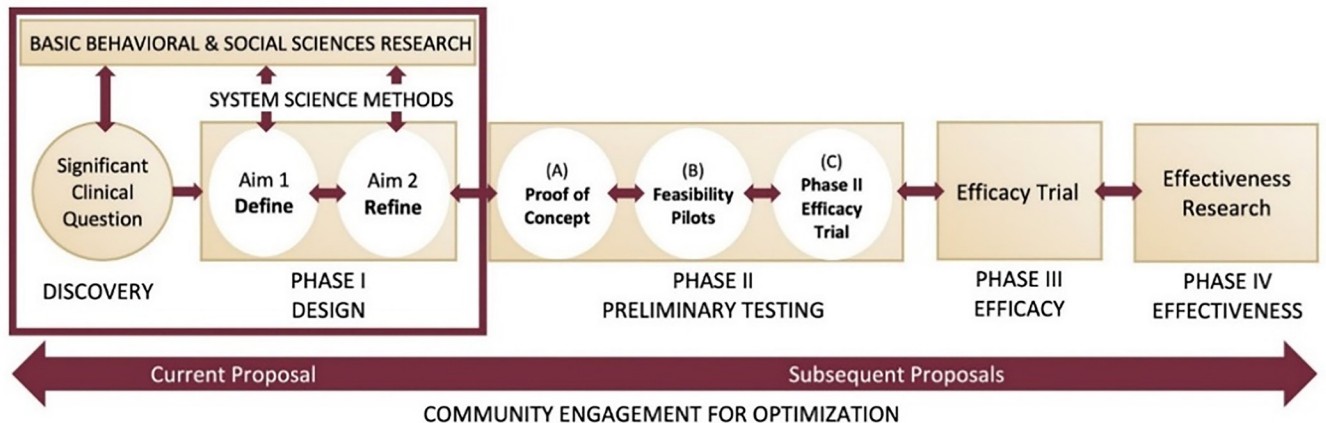

**Fig 1. ORBIT model as applied to developing antiracism interventions.**

development approach with associated research questions and methods. ORBIT Phase 1, the design phase, includes two subphases, *Define and Refine*. The following activities occur in *Phase 1a-Define*: develop hypothesized pathways by which the intervention can solve an important clinical problem, provide scientific basis for behavioral milestones, and provide basic behavioral and social science basis for treatment components and their respective targets. In ORBIT *Phase 1b-Refine*, activities include identifying independent and interacting treatment strategies, optimizing and tailoring for specific populations or systems, and ensuring appropriate measures to assess effects in future trials. In this phase, our project will simulate the effects of such strategies on community-defined health equity outcomes to guide which interventions should move on for further testing and will develop methods for identified gaps in outcomes measurement of such trials. The ORBIT model has typically been applied to behavioral interventions, and this will be the first project to apply the ORBIT model to multi-level interventions.

Third, we will use community-engaged systems science to identify critical leverage points for antiracism interventions in exemplary integrated primary settings in the Southern United States and use community and scientific advisors to develop possible intervention strategies. Systems science methods define critical leverage points ripe for disruptive innovations within complex systems [28]. A disruptive innovation in health systems is one that creates new networks and new organizational cultures to diffuse new practices to improve health outcomes and the value of health care [29]. Community-engaged systems science infuses Community-based Participatory Research practices to ensure that community partners are involved in systems mapping racism in health care, identification of leverage points as well as screening potential intervention strategies for community relevance [30].

Fourth, we leverage a highly specified community engagement model from the National Academy of Medicine to ensure rigorous and sustainable community engagement at all stages of the project (ref 28 from below) [31]. This model specifies measurable benchmarks for transition between phases of community engagement.

Finally, item response theory (IRT) provides a rigorous approach to initial measurement development critical for future intervention studies [32]. IRT models have distinct advantages over measures developed using classical test theory [31], which only provides a single estimate of average reliability. IRT demonstrates exactly when a test is giving more or less precise estimates of the attribute being measured. Additionally, IRT provides an estimate of "reliability" where single test can have a range of reliability depending upon how much of the attribute is present.

## Summary

Guided by the integration of these theoretical models, this protocol will develop and test a transformative community-engaged paradigm for developing antiracism interventions in integrated primary health care systems. We will focus on two primary health care systems in North Florida that provide integrated behavioral health within primary care. North Florida is considered the Deep South and its counties have he largest percentage of Black/African Americans in the state. Using community-engaged systems science methods, this protocol will map racism in provider behavior and organizational-systems practices, identify disruptive antiracism interventions, and simulate the impact of intervention models on provider behavior and organizational-systems' policies and practices. The end products will be the specification of full-scale clinical trials to test successfully simulated interventions and a paradigm.

## Materials and methods

The primary aims of the study are to: 1) map racism in provider behavior and organizational-systems practices toidentify target points for potential disruptive antiracism interventions (ORBIT Phase 1a); and 2) simulate the impact of potential antiracism strategies on provider behavior and organizational-systems' policies and practices on community prioritized outcomes (ORBIT Phase 1b). We integrate qualitative and quantitative systems science methods by using comumunity-engaged group model building with providers and Black/African American patients for aim 1 and computational simulation modeling with ongoing community input for aim 2. A secondary aim is to *i*nitiate the process of developing core measures to assess the effects of interventions at the provider and organizational-systems level. By using Item Response Theory, we will measure and test the relationship between disparities in healthcare latent factors and its manifestations, as specified in the primary aims.

## Proposed advisory councils

To effectively guide research practices and procedures as well as center the community throughout all stages of the process, the study will form several advisory groups encompassing varying levels of expertise. All advisors will be offered payment as consultants on the project at the same rate regardless of advisory group.

**Scientific Advisory Council (SAC).** This council will consist of experts on the prevalence of race-based differential treatment in health care, on community-based health equity research with Black/African Americans, and on interventions to reduce structural racism in primary care. The council of up to five members will meet as needed to provide project guidance, support and feedback on the process and scientific components of the project.

**Community Advisory Council (CAC).** This council of up to eight members will include those with expertise on healthcare systems in addition to the perspective of persons with experience with the two local health systems including providers, administrators, patients, and representative from the state Department of Health. Goals include interpreting data elicited from the project to help translate back to the study team and community constituents from a community lens. The group will also review dissemination products and informs sustainability plans. This council will meet approximately 3 times per year.

**Core modeling team to design GMB sessions.** This group is responsible for the design and convening of the GMB workshops. Each core modeling team will include project PIs and project leads as well as a representative from each health care system partner. Thus, there will be one core modeling building team for each of the two sites. This team will meet regularly in advance of the workshops to make specific design decisions and is central to facilitating the research activities for Aim 1 of the study. These site-specific teams will discuss substantive, methodological, logistical, and community perspectives of workshop design to ensure relevancy, effectiveness gather data to aid in problem specification.

**People's health justice committee.** In order include the voices of potential patients and community leaders from medically underserved populations who may not be fully engaged in primary care, another advisory group will be formed consistent with community-based participatory research approaches. The People's Health Justice Committee—a grassroots community group representing the Black community in the counties served by the two health system partners—will advocate for the experience and interests of residents in the community in a holistic way and will support shared power, trust, and sustainability. Committee members are considered experts of the community as well as critical sources of information to identify recognition of inequities and antiracism strategies in primary care. It is expected that this committee will convene 6 (1 hour) meetings per year.

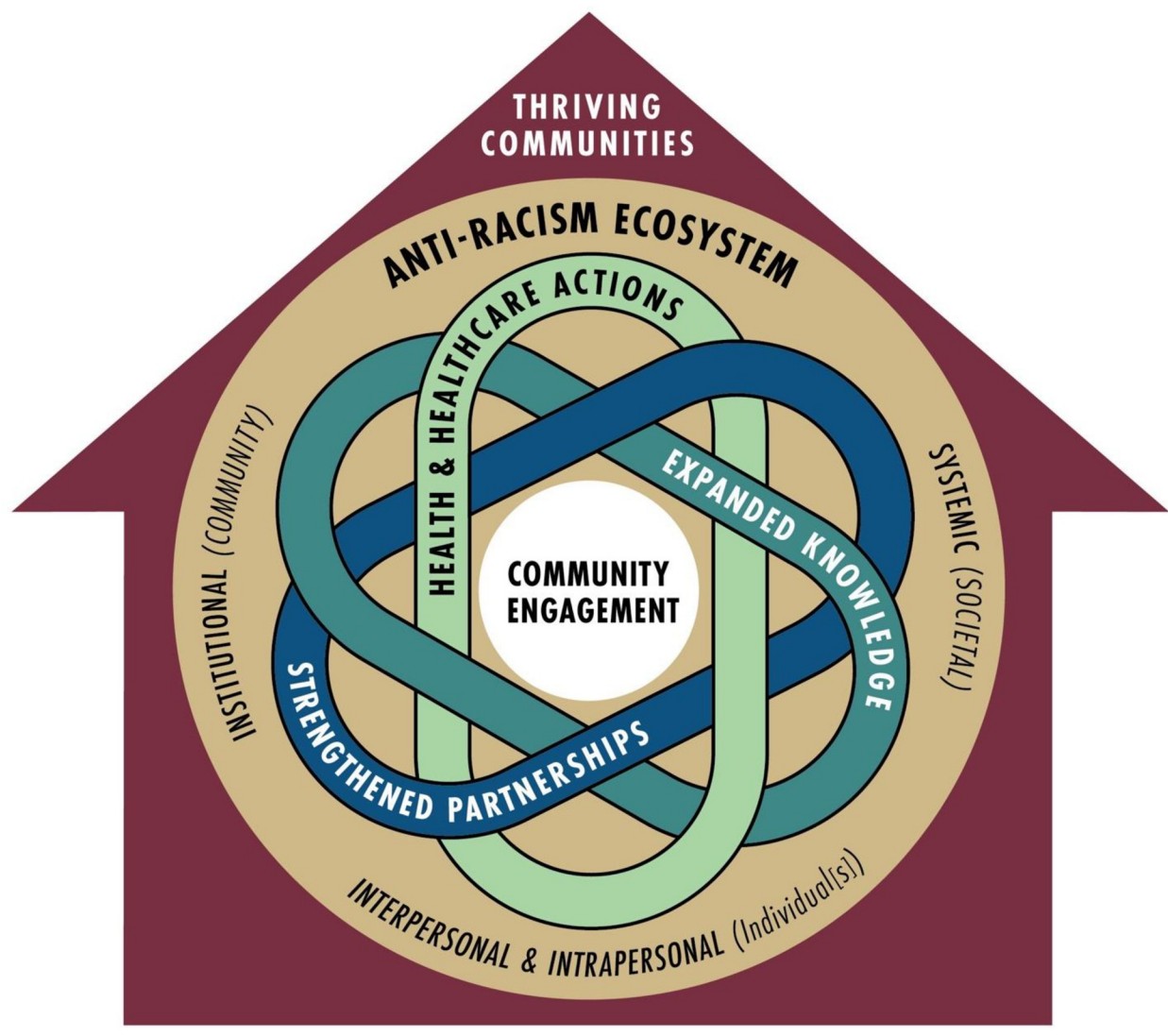

**Fig 2. Transformative health justice conceptual framework.**

### Transformative framework

A new transformative framework integrates the theoretical models described above (see Fig 2). At ORBIT Phase 0 we prioritize community engagement through planned information-

**Table 1. Benchmarks for Pillar 1 of the transformative health justice conceptual model.**

| DOMAIN | Benchmark |
|---|---|
| Diversity and Inclusivity | Perspectives reflect the composition of the community and the multidisciplinary expertise from the community (academic, healthcare systems, and broader community) (intentional integration of interests, knowledge, resources, perspectives). |
| Partnership and Opportunities | Participants benefitted from beginning to identify recognition of inequities and antiracism strategies, etc. |
| Acknowledgement, Visibility, Recognition | •Community members are seen and recognized as experts and as critical sources of information for problem framing and model building.<br>• Acknowledgement in dissemination products. Acknowledgements are approved by all participants. |
| Sustained Relationships | •Develop a collaborative sustainability plan beyond the 5-year period.<br>• Commitments to the sustainability plan from investigators and advisory groups. |
| Mutual Value | • Mutual problem definition and establishing system boundaries for study sites<br>• Consistent attendance at GMBs |
| Trust | •Contact / communication plan with study sites & community advisory groups.<br>• Train research team cultural humility, racism/anti-racism, and developing trustworthiness.<br>• Community advisory groups approve protocol for reviewing anything public facing |
| Shared Power | • Joint facilitation of community advisory group.<br>• Community advisory groups review dissemination materials and generated intervention strategies |
| Structural Supports for Community Engagement | • Manualized depiction of pillars reviewed by advisory groups<br>• Academic and community dissemination of results of each pillar<br>•Resources allocated to maintaining community advisory groups throughout the project and beyond |

gathering visits with primary care sites and community historical interviews and listening sessions before beginning the GMBs. This informs the ORBIT significant clinical question and moves into ORBIT Phase 1a (Define) by conducting the GMBs. Community engagement benchmarks will be completed for Pillar 1 Strengthening Partnerships before transitioning to Pillar 2 Expanded Knowledge which continues the Orbit Phase 1a (Define) by synthesizing GMBs and sharing them with community partners and advisory groups. Bidirectional communication in this pillar is critical to model refinement and subsequent simulations in ORBIT Phase 1b. Interventions specified from the simulations in collaboration with the advisory groups will address Pillar 3 Improved Health and Health Care Policies and Programs resulting in community friendly toolkits and future studies to test the real-world impact of these strategies that address Pillar 4 Thriving Communities. The social ecological model of antiracism ensures that multiple system levels are addressed simultaneously throughout the Define and Refine methods so that interventions across the ecosystem are considered.

To ensure authentic community engagement protocol activities tied to this framework, we formally adapted the Academy of Medicine's benchmarks for the current project (see Table 1 for Pillar 1 example) and benchmarks will be reviewed routinely by the investigators to determine readiness to transition between pillars.

## Description of GMB methods (Aim 1; Define Phase)

To address health equity through antiracism interventions, multiple stakeholders must play significant roles throughout the process. GMB is a community-engaged approach to inform

system dynamics models as an effective way to involve key stakeholders in the process of conceptualizing and formulating disruptive and innovative models for antiracism interventions [33,34]. GMB helps people visualize the interconnections and critique the process boundaries to support the development of innovative approaches to antiracism and health equity [35,36]. Drawing on earlier published work on GMB scripts and workshop techniques from participatory rural appraisal [33,37–39], community operations research [40], and soft systems methodology [41], the core modeling team will develop a set of GMB scripts to identify racism practices (provider and organizational-systems level) within and across the health care systems. GMB sessions will be organized around the use of "scripts"—structured group exercises designed to elicit information and engage participants. Each script defines the purpose of exercise, duration, preparation needed, activities, and outcomes.

Providers, administrators and patients will participate in separate workshops. Each workshop will aim to have no more than 15 participants. Systems science team members with training in system dynamics will function as recorders in the session, and one process coach familiar with group facilitation will be present. The GMB workshops will be organized around the modeling process moving from initial problem conceptualization, elicitation of key variables, defining the problem in terms of several key indicators and their behavior over time (reference modes), eliciting structural relationships, and formulating key relationships. GMB workshops will include a series of hands-on exercises designed to clarify key concepts in systems science (e.g., the distinction between stocks and flows and structure-behavior relationship). A final session will be convened separately for providers and patients to present back synthesized causal maps of structural drivers of racism practices as well as a provisional set of potential leverage points for antiracism interventions targeting provider behavior, and organizational-system policies and practices. Then participants will suggest any necessary revisions to the maps, review and prioritize possible intervention strategies based on their perception of impact, feasibility, and acceptability.

To promote antiracism in the GMB design, the core modeling teams will address how discussions on racist practices will be handled. In-depth discussion with participants will include common language and definitions surrounding racism at each level of the social ecological model and clarifications on implicit and explicit forms of discrimination. As GMBs proceed and there are mentions of persons of color that may have social economic limitations, for example, the GMB facilitator will ensures the use probing questions to understand perspectives on intersectionality and how multiple identities may or may not influence health outcomes. Script prompts will be intentionally framed to encourage reflections of racism at multiple levels and potential antiracism interventions.

## Sample size and recruitment

All participants (n ~ 100) will be over the age of 18 and members of one of the following categories for at least year: patients, providers, or agency/clinic/practice leaders. Participants will be excluded if they are assessed to have cognitive impairment that would interfere with their ability to participate. Patient stakeholders will be excluded if they do not self-identify as Black/African American. All participants receive a $50 gift card per workshop.

**Healthcare system participants.** In working with healthcare system liaisons, the research team will solicit recommendations of potential healthcare stakeholder participants representing each stakeholder group. Study team members will hold several in-person meetings with clinic representatives in order to build rapport and relationships, and to explain all aspects of the project, and to allow clinic stakeholders to ask any questions or share any concerns. Potential study participants will be contacted via email by the study site liaison at each site requesting

their participation and consented by a research team member prior to the GMB sessions. They will also be provided a study information sheet, detailing study information, elements of consent, and requesting to share their insight in the GMB workshops—inclusive of scheduled date/time for each stakeholder group. Participants will receive a $50 gift card for each workshop attended.

**Patient participants.** The content of GMB sessions, along with input from our advisory groups will inform how and who is recruited to participate in patient groups. Patients will be recruited using a variety of community-based strategies, including general community outreach activities, working with health system partners to advertise the study, and social media. Patients will be provided opportunities to express interest in participating by contacting the study team and/or direct screening and enrollment via online survey database. We will collect zip code data to consider the diversity of our sample as related to social disadvantage. Participants will receive a $50 gift card for each workshop attended.

## Description of simulation process (Aim 2; Refine Phase)

The following are the steps to achieve Aim 2 based on best practices [42] for building and testing system maps and computational models of complex social and organizational systems. Based on the Aim 1 GMB results and outputs, the first step will be to produce a set of systems maps (using causal loop diagrams) describing the important feedback loops and leverage points in the healthcare system. Depending on the form of these systems maps, the second step will be to develop a prototype simulation model of all or part of the healthcare provider system. If the various causal maps focus on higher-level aspects of the provider system, then the simulation may take the form of system dynamics modeling. For example, if the causal map suggests that there are patient intake chokepoints that drive patient population disparities, then a simulation model can examine the effect on patient equity via an intake expansion intervention [43].

Alternatively, if GMBs elicit critical microlevel system behaviors, agent based modeling may be appropriate. Hybrid models that, for example, utilize different modeling approaches for different parts of the systems maps are also possible [44]. The CBPR nature of the transformative framework precludes specifying which approach to use at the onset, and instead requires the careful analysis of the GMB results and the related insights of the advisory groups to determine the necessary simulation approaches.

For the simulation model to be applied, a computational architecture of the model will be established, and each piece will undergo testing to ensure appropriate representation of concepts and relationships. Initial model parameters will be revised during model development and testing in collaboration with the advisory groups. The generative explanatory power of the model is to reproduce observations about antiracism interventions at provider and organizational-systems levels to be tested under a variety of conditions. For example, across varying contextual factors and system characteristics, does role playing and coaching change provider behavior? Or do strategies to change organizational culture have the same level of influence on antiracism practices across context factors? These findings will be used to calibrate the model, establish the parameters for sensitivity analysis, and finalize the model.

There is a dearth of data on antiracism in primary care and our team has the ability to conduct simulations in the absence of such data. In fact, some of the most influential systems science models were developed without data [45,46]. To study phenomena and systems where little empirical data exist (i.e., antiracism in primary care) *prospective* computational modeling can work not only to identify leverage points in the system where interventions may perform best, but also to illuminate data gaps and which metrics and specific variables are most needed to increase the evidence

base [47]. Developing a community- informed simulation model, identifying leverage points for interventions, and subsequently developing interventions for antiracism in primary care will not only advance the field in general, but also expose critical data gaps along the way forming recommendations for the field and best data collection practices [48].

In addition to the systems maps and models, we will produce a set of translational resources that can help communicate the systems knowledge generated with our provider and patient communities. In particular, we will develop an interactive dashboard of model results via the R package Shiny [49], which will allow for different types of partners to experiment with intervention levels and implementation strategies firsthand and help to increase engagement with model and project results.

## Measurement considerations (secondary aim)

During this simulation phase, we will identify measurement gaps both for future modeling and for intervention testing. In addition, through expert consensus of investigators and advisory groups, we will determine if the measurement foci are single latent constructs made up of dependent items or a summation of independent components. That is, are the concepts we need to measure latent, reflective indicators (e.g., mental health) or formative indicators (e.g., life events)? [50] Subsequently, we will consider item development and explore possible methods including self-report surveys, interviews, health records, and mystery shopper approaches.

## Ethical considerations and declarations

This protocol has been approved by Florida State University Institutional Review Board on May 6, 2022 (Study 00003039). Human subject research will take place in Phase 1 of the protocol, which encompasses stakeholder (organization leaders, providers, and patients) recruitment and participation in GMB workshops (Primary Aim 1) with informed consent. The study will take several steps to minimize risks of participation to stakeholder participants. We will hold several GMB workshops that encompass different categories of stakeholders. Stakeholders will only be in GMB workshops with stakeholders in the same category. For example, there will be GMB workshops for patients only, that will only host stakeholders that are patients of the healthcare system. Healthcare providers or any professional affiliated with the healthcare system will NOT be present during the patient GMB workshops, nor will they be made aware of any comments made by patients or other stakeholders. This is also applicable to frontline workers such as physicians, residents, nurses and social workers and healthcare administrators. These groups will be having GMB workshops separately to protect confidentiality and allow open discussions without fear of adverse consequences such as retaliation. Furthermore, outcomes of data collection from the GMB workshops will not be individually reported. Due to the nature of GMB, individualized comments and discussions will not be reported. Data collection outcomes will be used to develop system mapping tools and causal loop diagrams.

Partners will be assured that their information will be kept confidential to the best of the study team's ability, and the responses they provide will have no impact on their relationship with any academic or health institution. GMB workshops will take place in a private space where stakeholders can openly express themselves. Additionally, the GMB workshop facilitation team has had extensive training and experience in conducting GMB workshops, which includes training on navigating sensitive topics in qualitative research. If stakeholder participants have any concerns about any aspect of the study, we will make it clear that they may choose to stop participating at any time without penalty. Participant stakeholders are under no obligation to participate, nor will their employment or medical care be compromised because based on their decision to participate.

A critical ethical consideration is to ensure that antiracism is infused through the research team's policies and practices. To that end the following procedures will be put in place: 1) Ensure diversity of project staff and GMB facilitators; 2) Ensure that all investigators have documented training in racism/antiracism (e.g., through university, NIH); 3) Follow guidelines for inclusivity in communication; [51] 3) Follow guidelines for antiracism in the implementation and disseminating of research; [52] 4) Plan for ongoing discussion of antiracism practices in data science [53,54].

## Community-engaged dissemination plans

Aligned with our conceptual framework, dissemination of protocol findings and lessons learned will prioritize community involvement, shared power, trust, community ready information, acknowledgement, visibility and recognition. The community will be involved in all dissemination plans by utilizing the multidisciplinary expertise of our People's Committee and CAC. Community-engaged dissemination planning will incorporate a scope to yield one dissemination product for each pillar in our conceptual model. Dissemination plans for each pillar will be discussed among community groups to get insight on type of dissemination products—alternatives to peer review journals—tailoring and framing messaging for specific audiences, and feasible timeframes to allow for sufficient review and feedback of materials. Community groups will be asked to approve any dissemination material or recommend changes for anything public facing. Furthermore, we will prioritize acknowledgement of community in dissemination products, and acknowledgements will be approved by all collaborating community members (e.g., CAC and People's Committee).

## Timeline

Fig 3 shows the project timeline. The transformative program requires a planned and an alternative timeline based on potential problems and alternative plans, and below demonstrates the management of potential timeline disruptions.

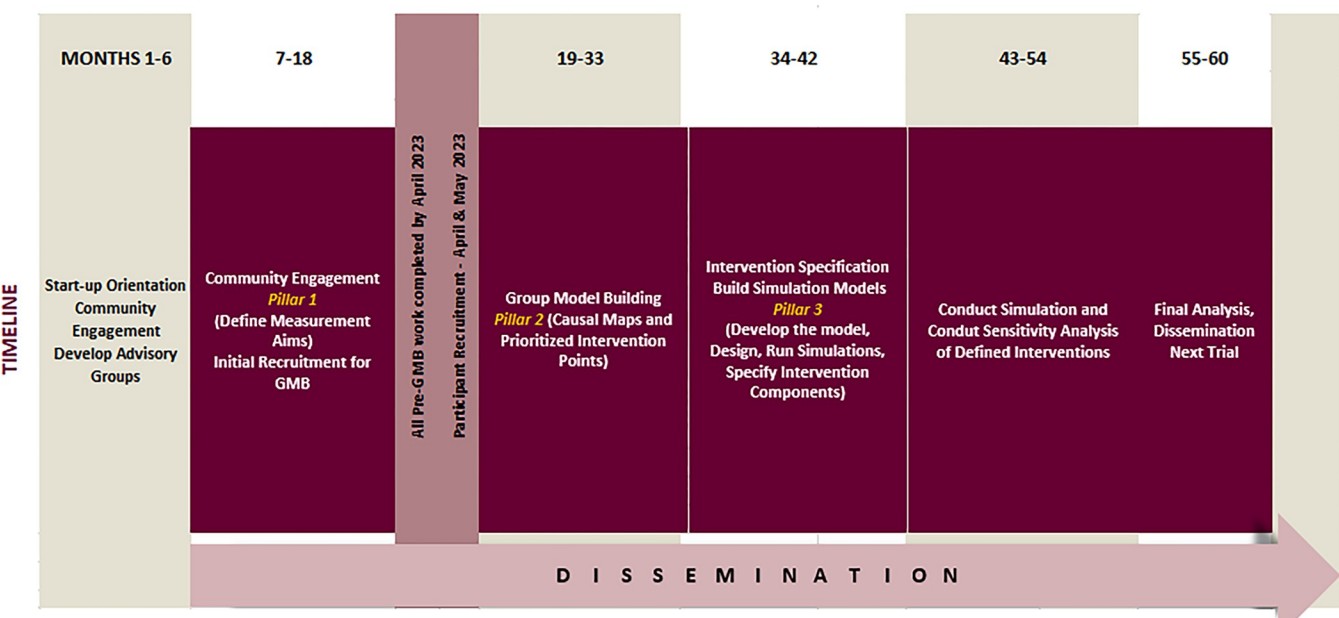

**Fig 3. Project timeline.**

## Discussion

Despite decades of studies acknowledging racial health disparities and an increased awareness of the social determinants of health, we seem to be light-years away from significant change. Racism still very much impacts the health care delivery system even in systems designed for low income and minority populations that strive to integrate physical and mental health treatment in primary care [55]. As noted in *Lancet*, racism kills [56], and there are insufficient evidence-based antiracism interventions across the antiracism ecosystem. Beyond one-time, brief training modules, which have minimal impact on behavior [21] little has been done to combat racism and discrimination in the context of primary care in the US. We propose to develop and test a transformative paradigm that integrates the ORBIT model for intervention development, systems science approaches, and a community engagement measurement framework for early phase translation of basic behavioral and social science into new antiracism interventions in integrated primary care settings that serve Black/African American patients. The paradigm is the first of its kind to integrate a social ecological model of racism, community-based participatory research, systems science, and community engagement model with benchmarks with in a phased model of translational science. Potential limitations include lack of consensus on systems maps resulting from GMB workshops and the need to conduct simulations with limited available data. Modeling portions of the system for possible intervention strategies and updating the models in the future with emerging data may be necessary. Furthermore, we do not expect that the findings will translate to other marginalized populations experiencing racism, but that the approach could be applied to antiracism strategies in other contexts.

The paradigm leverages an established framework of early phase translational behavioral and social science to rigorously define and refine new multilevel antiracism interventions within complex health systems and begin to rigorously develop measures to assess impact with community engagement formalized at every step. Without such a transformative paradigm, antiracism provider and organizational intervention research will remain stagnant and have minimal impact on redressing mental and physical health inequities because of such a fragile evidence-base.

The primary lessons learned to date are rooted in authentic community engagement. First, we contend that project activities build "racial stamina" [57] and bolsters the ability to discuss racism across stakeholder groups. For patients, the focus on antiracism ensures a commitment to strategy developing and implementation as opposed to ongoing documentation of racism in health care with little change.

Second, it is critical to not only develop a community-engaged framework, but also to operationalize associated benchmarks to ensure authenticity. This can be new an investigator who is accustomed to following a prescribed timeline of study activities, but is critical to avoid premature movement across translational research phases. It is clear that sufficient time must be allocated to authentic community engagement in the first pillar of framework activities (Strengthening Partnerships) before initiating data collection activities. The addition of community GMBs in addition to provider and patient GMBs may further support community engagement in Pillar 2. Because the ORBIT model has not specified community engagement activities, the integration of ORBIT with community engagement pillars further enhances this model of translational behavioral and social science and extends the model to multilevel interventions.

Third, in addition to initiating GMB workshops followed by simulation activities and measurement objectives, part of the new transformative health justice framework focuses on developing sustainable solutions and what determining residual infrastructure and community resources are needed to further refine, test and implement solutions. When investigators have

multiple demands, projects, and research interests, it is critical to determine up front who is committed to investing time and resources to lead initiatives beyond the project period. Our benchmarks include the development of an initial sustainability plan by the end of year 2 of the project and determine which investigators will collaborate with community members on testing and implementing identified antiracism strategies to reduce health disparities in integrated primary care in North Florida. By establishing a process of rigorous translational research in community-engaged antiracism strategy development and testing, this transformative project will produce a replicable, innovative, efficient and effective community-engaged framework for addressing health care racism and transform efforts to improve integrated primary health and mental health care services. Leveraging this transformative paradigm to disrupt systems that are producing health inequities will not only improve health outcomes for the current target community but may be reproducible to improve other communities and other health delivery systems.

## Acknowledgments

The authors would like to acknowledge our collaborating health system partners, community residents, and advisors who shared their insight and expertise on health equity, healthcare systems, and community engagement and helped shape and refine project methods for optimal success in transforming health equity. This work is supported by the NIH Director's Transformative Research Award Grant No. 5R01MD017404-02. The contents of this paper are solely the responsibility of the authors and do not necessarily represent the official views of the NIH.

## Author Contributions

**Conceptualization:** Sylvie Naar, Carrie Pettus, Norman Anderson, Meardith Pooler-Burgess, Penny Ralston, Heather Flynn, Todd Combs, Christopher Schatschneider, Douglas Luke.

**Data curation:** Sylvie Naar, Carrie Pettus, Meardith Pooler-Burgess, Penny Ralston.

**Formal analysis:** Todd Combs, Christopher Schatschneider, Douglas Luke.

**Funding acquisition:** Sylvie Naar, Carrie Pettus, Norman Anderson, Heather Flynn, Douglas Luke.

**Investigation:** Sylvie Naar, Carrie Pettus, Meardith Pooler-Burgess, Claudia Baquet.

**Methodology:** Sylvie Naar, Carrie Pettus, Norman Anderson, Penny Ralston, Heather Flynn, Todd Combs, Claudia Baquet, Christopher Schatschneider, Douglas Luke.

**Project administration:** Sylvie Naar, Meardith Pooler-Burgess.

**Supervision:** Sylvie Naar, Carrie Pettus, Norman Anderson, Meardith Pooler-Burgess.

**Writing – original draft:** Sylvie Naar.

**Writing – review & editing:** Sylvie Naar, Carrie Pettus, Norman Anderson, Meardith Pooler-Burgess, Penny Ralston, Heather Flynn, Todd Combs, Claudia Baquet, Christopher Schatschneider.

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
