## [Decision Letter · Decision Letter 0]

4 Jan 2024

PONE-D-23-32179Study Protocol for Transforming Health Equity Research in Integrated Primary Care: Antiracism as a Disruptive InnovationPLOS ONE

Dear Dr. Naar,

Thank you for submitting your manuscript to PLOS ONE. After careful consideration, we feel that it has merit but does not fully meet PLOS ONE’s publication criteria as it currently stands. Therefore, we invite you to submit a revised version of the manuscript that addresses the points raised during the review process.

We look forward to receiving your revised manuscript.

Kind regards,

Sze Yan Liu, PhD

Academic Editor

PLOS ONE

Additional Editor Comments:

This study may potentially yield important results. The study protocol describes an interesting idea. However, similar to the reviewers, I thought the protocol needs more clarity regarding the specific population and the role of the community.

Reviewers' comments:

Reviewer's Responses to Questions

**Comments to the Author**

1. Does the manuscript provide a valid rationale for the proposed study, with clearly identified and justified research questions?

Reviewer #1: Yes

Reviewer #2: Partly

Reviewer #3: Yes

2. Is the protocol technically sound and planned in a manner that will lead to a meaningful outcome and allow testing the stated hypotheses?

Reviewer #1: Partly

Reviewer #2: No

Reviewer #3: Yes

3. Is the methodology feasible and described in sufficient detail to allow the work to be replicable?

Reviewer #1: Yes

Reviewer #2: No

Reviewer #3: Yes

4. Have the authors described where all data underlying the findings will be made available when the study is complete?

Reviewer #1: No

Reviewer #2: No

Reviewer #3: Yes

5. Is the manuscript presented in an intelligible fashion and written in standard English?

Reviewer #1: Yes

Reviewer #2: Yes

Reviewer #3: Yes

6. Review Comments to the Author

You may also provide optional suggestions and comments to authors that they might find helpful in planning their study.

Reviewer #1: Please see attachment for comments. Overall, this is exciting research and I look forward to the study. The protocol would benefit from some reorganization and editing for clarity.

Reviewer #2: Manuscript No.: PONE-D-23-32179

Manuscript Title: Study Protocol for Transforming Health Equity Research in Integrated Primary Care: Antiracism as a Disruptive Innovation

Study purpose: This paper provides an update on a process to transform health equity research in integrated primary care. In addition to presenting the ORBIT project framework, the authors note how this framework was updated to authentically engage community members and develop benchmarks to ensure community members are better intentionally integrated into the project.

The authors have nicely laid out the lack of available and effective antiracism intervention in primary care. However, it will be important for the authors to be more explicit about what makes their approach antiracist and how this approach will address the various forms of racism that exist that are interrelated from cultural, systemic, interpersonal, and vicarious, or provide more specificity what aspect of racism is being targeted and why.

As currently laid out, there is an assumption of linearity without attending to the endemic of racism and how it morphs and changes under various conditions, as laid out by Anderson et al. (2022). The authors should also interrogate the extent to which the ORBIT framework is antiracist from conceptualization to implementation. This lack of interrogation of frameworks, processes, and analytical approaches may still result in interventions that are racist from inception through implementation, whether intentional or unintentional. For example, how do the scripts of the group model building (GMB) ensure that both implicit and explicit examples of racist practices are called out, especially with attending to intersectionality? Similarly, what would be the process of identifying and selecting participants in the workshops, including facilitators, etc.?

As the authors recognized that more attention is needed to recruit and authentically engage vested community partners, what are the implications for the simulation process? To what extent is the simulation model taking a quantcrit approach (e.g. Gilborn et al., 2018)? For example, there is a discussion about holding constant contextual factors as part of sensitive analyses (p.13, line 312). When racism is endemic to the various ways health care systems and the U.S. function, how does holding contextual factors constant aligned with the understanding of the various forms of racism?

It is still unclear what the potential intervention will be and what will be changed and who will benefit. What are the potential interventions and potential outcomes, and at what level is change expected? If the ORBIT approach, which is foundational to this paradigm shift proposed in this project, does not even consider community engagement, there is a concern about whether this approach will actually get to the expected outcomes.

On a relatively minor note, I would advise that the authors consider some of their language, including the use of the term stakeholders, which the Centers for Disease Control and Prevention (2023) said “has a violent connotation for some tribes and tribal members.”

I appreciate the authors’ discussion, but trying to understand how their results align with antiracism which could lead to vast changes considering it is unclear to what extent the community engagement will be weaved throughout all aspects of the project, especially in the measurement and simulation aspects.

Anderson, R. E., Heard-Garris, N., & DeLapp, R. C. T. (2021). Future Directions for Vaccinating Children against the American Endemic: Treating Racism as a Virus. Journal of Clinical Child & Adolescent Psychology, 1-16. https://doi.org/10.1080/15374416.2021.1969940

Centers for Disease Control and Prevention (2023). Health Equity Guiding Principles for Inclusive Communication. Washington, DC: Author. https://www.cdc.gov/healthcommunication/Health_Equity.html

Gillborn, D., Warmington, P., & Demack, S. (2018). QuantCrit: education, policy, ‘Big Data’ and principles for a critical race theory of statistics. Race Ethnicity and Education, 21(2), 158-179. https://doi.org/10.1080/13613324.2017.1377417

Reviewer #3: Abstract:

It took me a while to realize that this study is to assess Racism in the African American (AA) population.

Not only is it not mentioned in the Abstract, but it also needs to be clarified in the Intro (see my comments below). It is essential to be intentional about describing in the Introduction why you are doing this system-wide intervention and measuring it only amongst AA (leaving all the other rich Florida diversity).

Introduction:

Having the first phrases describing only AA statistics is confusing, particularly after the Abstract does not mention that you will only look at the AA community. I recommend starting with a broad description of the issue that affects the global majority (those racialized and historically marginalized communities, formerly known as minorities). Being that the funding goes to researchers in Florida, I am surprised that there are no reports of disparities amongst Latine and Immigrant communities. Make the case why, and probably given the need to involve the community; funding is limited to do this work in a participatory way with one. Build the case for that.

It is important to clarify that there are disparities in health co-morbidities and mental health due to Structural Issues (social determinants of health), which create the main difference between communities. Thus, many Community-based, policy-based structural interventions exist to stop that. Then there is a different set of differential treatments that happens when a patient comes (if they can) to our system: unequal treatment. This is what you are referring to, and this is what is essential to highlight. Even when we put office-based interventions to buffer some of the SDOH impacts, it will differ from the band-aid we are creating locally. Ensure some of this goes into the Introduction, explaining the disparities' origin. See the figure 1 on your citation number 1: Unequal treatment ( I tried to paste it here but could not).

In the Introduction, I advise discussing concepts about Intersectionality and how they play a role in the pile of discrimination and how we need to account for that when talking about anti-oppression. There are several articles defining the need for this. One has coined the name of Integrative racism (Nelson, L.E., Reeves, J. and Lopez, D.J., 2023. Integrative Antiracism and the Salience of Intersectional Assets for Black and Latinx LGBTQ Youth. Journal of Adolescent Health, 72(5), pp.647-648.), to signal that we need to pay attention to the interaction of other marginalized identities when designing anti-oppressive interventions, even when we are trying to tackle the forgotten one: racism.

Line 161: Add African American to "Community-engaged Group Model Building (GMB)"

Line 220: are the stakeholders paid? A genuinely participatory community-based study pays the stakeholders, patients, and providers for their involvement and time. I hope you do, and if so, add it to the protocol. Explain how you are handling the payment (cards?). Same with CAC: are they paid?

Line 247 Define MPIs.

Line 291: you could add some neighborhood-level data as part of contextual data when selecting patients, as this needed to be included to "diversify" patients. Examples below:

Line 338: Hurray! I was going to ask why the Intro did not talk about CBPR, as this was what you were trying to do in the initial phase; I'm glad you figured that out.

Line 346: paid? This is one of the most essential parts of a participatory process. If you are not doing that now, re-think how you could do that.

349: Excellent!

Overall result-discussion: I will probably eliminate the section Results (you mentioned before in the abstract that you would not share results, which are not results). I would incorporate this into the Protocol in Orbit phase 0. You could clarify there that this was not an initial component of the Funding (it seems like it was not), but for readers to understand the protocol, I will include it there, with a little summary on how you learned the importance. At this point in science, it is well established that this participatory component is one of the most important in Health Equity science and probably was an oversight on your initiative. I will add a paragraph about CBPR and discuss why. It seems more straightforward and accessible for the reader for protocol publishing purposes. This is my recommendation, of course, it is up to you to decide. Then, you should leave the discussion as it is.

I will add somewhere in the translational piece that the researchers will engage with the Site Leadership to start working on their buy-in to incorporate the suggested lessons learned during this study. Leadership buy-in is one of the most critical components for sustainability and health equity transformation. Thanks for the fantastic work and effort to advance health equity!

Thanks again.

7. PLOS authors have the option to publish the peer review history of their article (what does this mean?). If published, this will include your full peer review and any attached files.

Reviewer #1: No

Reviewer #2: No

Reviewer #3: **Yes: **Maria Veronica Svetaz

---

## [Author Response · Author response to Decision Letter 0]

13 May 2024

Reviewer 1

Major comments: 

1) Line 172: The paper goes directly from aims to the ethical considerations section, which mentions a lot of activities (e.g., GMB sessions). It seems like a description of these activities directly following the aims and prior to the ethical considerations section would be helpful for the reader. The advisory councils should be introduced here.

Response: We moved the ethical considerations paragraph to the end of the methods section and before the timeline, which is more consistent with the PLOS protocol template. The advisory councils are now introduced first. 

2) I wonder if the participants / recruitment section should go after the description of group modeling (starting line 255), so the reader knows what the recruitment is for. 

Response: We have reorganized accordingly. 

3) Line 289: Until this mention of agents, I had assumed that your reference to computational simulation modeling referred to system dynamics modeling. But it seems like you’re using agent-based simulation? I think this should be more clearly stated earlier in the methods (and likely the intro and abstract). Using ABM alongside systems mapping in a GMB process is not as common as SD modeling and represents a unique aspect of this study. 

Response: We have edited the methods to note that we did not pre-specify modeling approaches to be more consistent with a community-engaged framework and allow the GMB findings to dictate the approach to simulation. 

The following are the steps to achieve Aim 2(based on best practices) for building and testing system maps and computational models of complex social and organizational systems. Based on the Aim 1 GMB results and outputs, the first step will be to produce a set of systems maps (using causal loop diagrams) describing the important feedback loops and leverage points in the healthcare system. Depending on the form of these systems maps, the second step will be to develop a prototype simulation model of all or part of the healthcare provider system. If the various causal maps focus on higher-level aspects of the provider system, then the simulation may take the form of system dynamics modeling. For example, if the causal map suggests that there are patient intake chokepoints that drive patient population disparities, then a simulation model can examine the effect on patient equity via an intake expansion intervention. 

Alternatively, if GMBs elicit critical microlevel system behaviors, agent based modeling may be appropriate. Hybrid models that, for example, utilize different modeling approaches for different parts of the systems maps are also possible. The CBPR nature of the transformative framework precludes specifying which approach to use at the onset, and instead requires the careful analysis of the GMB results and the related insights of the advisory groups to determine the necessary simulation approaches. 

4) I was surprised to see a results section in a protocol paper, and also a bit confused about its contents. Please see minor comments below for questions re: timing, etc. I think you have a judgement call to make regarding whether the protocol should describe your original plans only or whether you’d like to include the additions / changes you made that are described now in the results. If you keep the additions / changes, I’m wondering if it would be clearer for the reader if the protocol was written as if all activities were proposed and would happen in the future. The detail about changing the stakeholder groups and engagement due to adapting to feedback could be detailed in future publications when communicating study results. But this is ultimately up to the authors and the journal editors. If the results section is kept, I’d recommend adding some explanatory text earlier so the reader understands what to expect regarding timing and changes. I feel like I’m not communicating this concern very well – please feel free to reach out if it’s not clear. 

Response: This comment makes a lot of sense! We removed the results section to more carefully match the PLOS One template. We now consider the methods changes to be part of protocol refinement as they occurred before recruitment was initiated and present them as part of the protocol. We removed the community GMB additions as this was not originally proposed, and recruitment procedures are not yet specified. 

5) Line 405: Somewhere, the paper would benefit from a clearer description of these theories and how they were integrated into your approach. Also, should item response theory be included? 

Response: We added a new section in the introduction to review each of four approaches that we integrate into a transformative framework.

6) Discussion: A few questions I’d like to see addressed in the discussion (or somewhere in the text): What are the mechanisms by which the modeling activities will lead to change for clinics and patients? Are you expecting participants to change their own behavior or the policies / procedures at their clinics or in their communities? How will you know change has happened? 

Response: We are not hypothesizing that the modeling activities in aim 1 will lead to change per se. Rather, the modeling activities will create a systems map to identify potential intervention points for antiracism strategies. Future studies will then test community prioritized strategies that have the greatest effect on the system based on simulations in aim 2.

7) Engaging in modeling is time intensive. How will you ensure that this experience is worthwhile for participants and not an additional burden? Is it safe to assume that people who participate are likely to be people concerned about addressing racism? If that’s the case, how will you avoid ‘preaching to the choir’ to ensure that structural changes actually get identified / made? 

Response: GMB participants will receive compensation for their time as research participants. All providers at the sites are recruited for GMBs regardless of their motivation for antiracism. Community advisors are paid consultants and we will assess their experience of engagement and mutual value as part of the benchmarks described in Table 1. 

8) Simulation requires data or assumptions to provide a foundation. I’m guessing that this model will require a lot of expert judgement due to a lack of extensive data. Building simulation models based on expert / stakeholder judgement is a valid approach, but necessarily involves challenges regarding how to elicit and combine perspectives into a single model. I’d like to see the authors address how they’ll approach these challenges in the methods and then discuss the broader implications (e.g., validity, generalizability, etc.) in the discussion. 

Response: We added the following to the methods and discussion sections (with references). 

Methods; There is a dearth of data on antiracism in primary care and our team has the ability to conduct simulations in the absence of such data. In fact, some of the most influential systems science models were developed without data. To study phenomena and systems where little empirical data exist (i.e., antiracism in primary care) prospective computational modeling can work not only to identify leverage points in the system where interventions may perform best, but also to illuminate data gaps and which metrics and specific variables are most needed to increase the evidence base. Developing a -community-informed simulation model, identifying leverage points for interventions, and subsequently developing interventions for antiracism in primary care will not only advance the field in general, but also expose critical data gaps along the way forming recommendations for the field and best data collection practices. 

Discussion: Potential limitations include lack of consensus on systems maps resulting from GMB workshops and the need to conduct simulations with limited available data. Modeling portions of the system for possible intervention strategies and updating the models in the future with emerging data may be necessary. 

Minor comments: 

9) The paper would benefit from minor proofreading corrections and grammatical edits (e.g., in line 71, ‘is’ should be ‘are’; the quotation starting in line 77 is not ended). 

Response: We apologize and have edited accordingly. 

10) Line 127: Is ORBIT an acronym? If so, spell it out. This section would also benefit from more description of the ORBIT model (e.g., how many phases does it have, is it iterative, what purpose does it serve, etc.) prior to discussing the individual phases. A figure or table summarizing this model might also be helpful. Line 147: I recommend rewording the reference to the figure to make it clear that the figure shows the protocol, which draws from the ORBIT model. The text now seems to say that the figure summarizes the established model. 

Response: We explicate ORBIT in the new theoretical models section. We have a figure of ORBIT and the parts that will be in this protocol versus future studies. We name this figure ORBIT as Applied to Developing Antiracism Interventions.

11) Line 157: I recommend having some kind of introductory text at the beginning of the methods giving a broad summary of the methods to be used, and maybe some introductory text prior to the list of methods to set them up. I also recommend either rewording the list of aims so that they are in full sentences or reformatting into a table with aim, approach, etc. 

Response: We have revised accordingly.

12) Line 169: Either here or earlier, a description of item response theory would be helpful. 

Response: We have added this to the theoretical models section in the introduction.

13) Line 201: At first I thought this section missed info about recruitment, but then I saw the following subsections. You might want to have a sentence in this first section that refers to different recruitment strategies for different stakeholder types to cue up the subsections. Also, are participants assembled into an expert panel? Are they reimbursed? How will it be determined whether participants identify as Black/African American? 

Response: We have clarified the different recruitment strategies for providers and patients in different sections with clear headers.

14) Line 203: Be consistent about past vs future tense throughout. 

Response: We have edited to primarily utilize future tense .

15) Line 223: How will patients be contacted? 

Response: Our patient recruitment section now shows that patients will be recruited using a variety of community-based strategies, including general community outreach activities, working with health system partners to advertise the study, and social media. Patients will be provided opportunities to express interest in participating by contacting the study team and/or direct screening and enrollment via online survey database.

16) Line 231: I’m a little confused about the different groups. You refer to recruiting participants, having community advisory groups (line 225) and now scientific advisory councils. A high level overview of the different groups or ways that participants interact with this research would be helpful, preferably at the beginning of the methods section. 

Response: We have reorganized significantly. The advisory groups are described first, and our paid as consultants. Recruitment of participants is described separately, and the gift card compensation for provider and patient participants is more clearly described .

17) Line 255 section: Will patients and stakeholders from other groups participate in causal mapping together or separately? How will you handle potentially different perspectives between individuals and groups? How are modeling decisions made? Will racism and other key concepts be defined previously or defined by the group(s)? 

Response; We added a statement to clarify that providers and patients will participate in separate GMBs. We note in the discussion that lack of consensus is a potential limitation. While we will attempt to integrate, and if necessary, we may have to model different perspectives separately. 

We have also added the following to the GMB description: To promote antiracism in the GMB design, the core modeling teams will address how discussions on racist practices will be handled. In-depth discussion with participants will include common language and definitions surrounding racism at each level of the social ecological model and clarifications on implicit and explicit forms of discrimination. As GMBs proceed and there are mentions of persons of color that may have social economic limitations, for example, the GMB facilitator will ensures the use probing questions to understand perspectives on intersectionality and how multiple identities may or may not influence health outcomes. Script prompts will be intentionally framed to encourage reflections of racism at multiple levels and potential antiracism interventions.

18) Line 283 Simulation process description: How will stakeholders be involved in operationalizing the simulation model? Will they help choose key indicator variables or have those been chosen (and if so, how?)? Will there be some iteration between model drafts and participant feedback? 

Response: We added a statement to the methods explaining the iteration between model drafts and advisory group feedback. 

For GMBs: A final session will be convened separately with providers and patients to present back synthesized causal maps of structural drivers of racism practices as well as a provisional set of potential leverage points for antiracism interventions targeting provider behavior, and organizational-system policies and practices. Then participants will suggest any necessary revisions to the maps, review and prioritize possible intervention strategies based on their perception of impact, feasibility, and acceptability. 

We added three statements to the methods for computational modeling: 

a) The CBPR nature of the transformative framework precludes specifying which approach to use at the onset, and instead requires the careful analysis of the GMB results and the related insights of the advisory groups to determine the necessary simulation approaches. 

b) Initial model parameters will be revised during model development and testing in collaboration with the advisory groups.

c) In addition to the systems maps and models, we will produce a set of translational resources that can help communicate the systems knowledge generated with our provider and patient communities. In particular, we will develop an interactive dashboard of model results via the R package Shiny(42), which will allow for different types of partners to experiment with intervention levels and implementation strategies firsthand and help to increase engagement with model and project results.

19) Will any quantitative data be used as a basis for model parameterization or will it be based solely on expert opinion? 

Response: See response to point 7 above. 

20) Line 332: Which initial activities? I had thought this protocol proposed future activities? Or are you referring to preliminary research that served as a basis for this proposed research? 

Response: Reference to initial activities has been removed. 

21) Line 336: Similarly, I’m wondering here about the timeline – Did recruitment already happen for these councils? It might be simpler for the reader if the protocol was worded in the future tense and the people’s health justice committee is described alongside the other councils. 

Response: We have revised accordingly.

22) Figure 2: The text in the figure is very small and will be difficult to read in the article pdf. Can this be revised to be more legible? 

Response: We have created a new figure that is more readable. 

23) Line 428: The community engaged dissemination plans should be better described prior to the discussion. 

Response: We have moved this section to the end of the methods. 

Reviewer 2: 

1) The authors have nicely laid out the lack of available and effective antiracism intervention in primary care. However, it will be important for the authors to be more explicit about what makes their

---

## [Decision Letter · Decision Letter 1]

12 Jun 2024

Study Protocol for Transforming Health Equity Research in Integrated Primary Care: Antiracism as a Disruptive Innovation

PONE-D-23-32179R1

Dear Dr. Naar,

We’re pleased to inform you that your manuscript has been judged scientifically suitable for publication and will be formally accepted for publication once it meets all outstanding technical requirements.

Kind regards,

Sze Yan Liu, PhD

Academic Editor

PLOS ONE

Additional Editor Comments (optional):

Your reorganization and additions have addressed the reviewers' comments.

Reviewers' comments:

Reviewer's Responses to Questions

**Comments to the Author**

1. Does the manuscript provide a valid rationale for the proposed study, with clearly identified and justified research questions?

Reviewer #1: Yes

2. Is the protocol technically sound and planned in a manner that will lead to a meaningful outcome and allow testing the stated hypotheses?

Reviewer #1: Yes

3. Is the methodology feasible and described in sufficient detail to allow the work to be replicable?

Reviewer #1: Yes

4. Have the authors described where all data underlying the findings will be made available when the study is complete?

Reviewer #1: Yes

5. Is the manuscript presented in an intelligible fashion and written in standard English?

Reviewer #1: Yes

6. Review Comments to the Author

You may also provide optional suggestions and comments to authors that they might find helpful in planning their study.

Reviewer #1: Thank you for your thoughtful and comprehensive response to feedback. I believe this manuscript is ready for publication.

7. PLOS authors have the option to publish the peer review history of their article (what does this mean?). If published, this will include your full peer review and any attached files.

Reviewer #1: No

---

## [Editor Report · Acceptance letter]

19 Jun 2024

PONE-D-23-32179R1 

PLOS ONE

Dear Dr. Naar, 

I'm pleased to inform you that your manuscript has been deemed suitable for publication in PLOS ONE. Congratulations! Your manuscript is now being handed over to our production team.

Kind regards, 

on behalf of

Dr. Sze Yan Liu 

Academic Editor

PLOS ONE